

**Dominant Influence of Biomass Combustion and Cross-Border Transport on Nitrogen-Containing Organic**
**Compound Levels in the Southeastern Tibetan Plateau**
Meng Wang[1, 2], Qiyuan Wang[1, 3, *], Steven Sai Hang Ho[4], Jie Tian[1], Yong Zhang[1], Shun-cheng Lee[5, *], Junji Cao[6,
*]
[1]State Key Laboratory of Loess and Quaternary Geology, Institute of Earth Environment, Chinese Academy of
Sciences, Xi'an 710061, China
[2]Department of Civil and Environmental Engineering, The Hong Kong Polytechnic University, Hung Hom, Hong
Kong
[3]CAS Center for Excellence in Quaternary Science and Global Change, Xi'an 710061, China
[4]Division of Atmospheric Sciences, Desert Research Institute, Reno, NV89512, United States
[5]Function Hub, Thrust of Earth, Ocean and Atmospheric Sciences, The Hong Kong University of Science and
Technology (Guangzhou), 511400 Guangzhou, China
[6]Institute of Atmospheric Physics, Chinese Academy of Sciences, Beijing 100029, China
*Correspondence to*: Qiyuan Wang (wangqy@ieecas.cn), Shun-cheng Lee (shunchenglee@hkust-gz.edu.cn), and
Junji Cao (jjcao@mail.iap.ac.cn)



## Abstract

The Tibetan Plateau (TP) is highly susceptible to climate change and the nitrogen-containing organic compounds (NOCs) in fine particulate matter ($PM_{2.5}$) represent one of the large uncertainties in affecting the climate in high-altitude areas. Previous studies have shown that NOCs play a vital role in the nitrogen budget of $PM_{2.5}$. However, our understanding of the composition and sources of NOCs in $PM_{2.5}$, particularly in TP, is limited. Here, we aim to enhance our understanding of NOCs in the TP region by examining their identification, concentration levels, sources, and origins. We conducted field sampling at a regional background sampling site in Gaomeigu, in the southeastern margin of TP from March 11th to May 13th in 2017. The daily mass concentrations of NOCs ranged from 714.4 to 3887.1 ng m$^{-3}$, with an average of (2119.4 ± 875.0 ng m$^{-3}$) during the campaign. This average concentration was approximately 40% higher than that reported at a typical regional site in the North China Plain (NCP), highlighting a more significant presence of NOCs in the Tibetan area. Biomass burning and secondary sources were identified as the major contributors to total NOCs. This was further substantiated by a regional air quality model, which indicated that over 80% of the aerosol in the southeast of TP originated from neighboring countries. This study enhances our understanding of NOCs' contribution to $PM_{2.5}$ in TP and their potential impacts on the climate stability in high-altitude areas.

*Keywords*: Southeastern Tibetan Plateau, Nitrogen-containing organic compounds, Source apportionment, Receptor model,



## 1 Introduction

The Tibetan Plateau (TP), located near densely populated and industrialized regions, is particularly susceptible to climate change (Meng et al., 2013; Duo et al., 2015; Li et al., 2015; Yuan et al., 2016; Zhao et al., 2022). The dry season features prevalent natural forest fires and anthropogenic burning activities, such as the combustion of agricultural residues, leading to substantial emissions of atmospheric pollutants (Zhao et al., 2015; Ran et al., 2022; Arun et al., 2021). Consequently, aerosol concentrations in the TP, especially during the premonsoon period, have risen markedly (Han et al., 2020). Previous studies in the Tibetan region have mainly focused on carbonaceous organic aerosols (OA), with nitrogen-containing organic compounds (NOCs) garnering less focus (Zhang et al., 2020; Zhang et al., 2019; Chen et al., 2014). NOCs play an important role in modulating climate, primarily through their light absorption abilities which influence aerosol radiative effects (Li et al., 2023). These compounds actively contribute to the formation of new particulate matter and secondary organic aerosols (SOAs), affecting cloud properties and the Earth's energy balance (Lin et al., 2021; Yu et al., 2024). The anthropogenic augmentation of nitrogen emissions has notably disrupted the global nitrogen cycle, with NOC deposition emerging as a significant source of reactive nitrogen (Li et al., 2023). This has profound implications for atmospheric chemistry and climate, necessitating a deeper understanding of NOC sources and atmospheric processes in the climate-sensitive region of TP.

The pre-monsoon period features meteorological conditions that facilitate the long-range transport of NOC-containing aerosols onto the TP, with prevailing atmospheric circulations transporting pollutants from neighboring countries in southwest China (Wang et al., 2019a). Anthropogenic biomass burning is more intensive during the pre-monsoon period and the incoming NOCs associated with biomass burning may have the potential to alter the chemical composition of the atmosphere, influence cloud microphysics, and affect the regional radiative balance during a critical time of hydrological accumulation and ecological transition (Tan et al., 2021). Given the TP's significance in the Asian water cycle and its role as a global climate regulator, the poorly characterized atmospheric behavior of NOCs during the pre-monsoon season represents a significant knowledge gap (Li et al., 2023).

Over the past decade, studies on NOCs have primarily focused on identifying their sources and concentrations (Song et al., 2017; Boreson et al., 2004; Barbaro et al., 2015; Lin et al., 2021). More than 200 NOCs have been detected in the atmosphere, originating from a variety of natural sources such as animals, vegetation, ocean, and husbandry, as well as anthropogenic sources including sewage treatment, combustion processes, vehicle emissions, and industrial activities (Zhu et al., 2020; Zhang and Anastasio, 2003b; Shi et al., 2010; Ho et al., 2019; Wang et al., 2022). Determining the sources of NOCs in the atmosphere remains challenging. For example, studies have identified the sources of specific NOCs like amines, amino acids, amides, nitriles, urea, and nitrophenol (Ge et al., 2011). Notably, Amines are prevalent in both urban and rural areas in America, mainly derived from industrial and animal husbandry (Sorooshian et al., 2008). Biomass burning and animal farming are known emission pathways for amino acids (Zhang and Anastasio, 2003a). Furthermore, investigations have shown that a significant portion of water-soluble organic nitrogen (WSON) may form secondarily, as indicated by its correlations with water-soluble ionic species like nitrate ($NO_3^-$), sulfate ($SO_4^{2-}$), and ammonia ($NH_4^+$) (Ho et al., 2015). Amides can react with atmospheric acidic particles, forming secondary aerosols (Priestley et al., 2018). Although previous studies have focused on identifying sources of prevalent NOCs (e.g., amino acids and amines) via tracer correlations, uncertainties about specific NOC concentrations and their sources persist. Recent studies have employed receptor models for source apportionment (Yu et al., 2024), yet a comprehensive understanding of NOCs is still lacking.

In this study, we collected fine particulate matter ($PM_{2.5}$) samples during the pre-monsoon season at a high-altitude, remote location near the Sino-Burmese border along the southeastern edge of the TP. The collected



samples were analyzed to determine their NOCs as well as carbonaceous components, water-soluble ions, and
elements. The objectives of the study were to investigate the general attributes and chemical composition of NOCs,
ascertain the contribution of various sources to these compounds, and identify the source regions influencing
$PM_{2.5}$ and specific chemical constituents in the area.
**2 Experimental**
**2.1 Sampling**
Aerosol sampling was conducted at the Lijiang Astronomical Station, the Chinese Academy of Sciences
(26.70°N,100.03°E, 3260 m above the sea level, Fig. S1) in Gaomeigu from March 11[th] to May 13[th] 2018. The
location is approximately 2 km away from the Gaomeigu village and 30 km from Lijiang City, located on the
southeastern edge of the TP (Zhao et al., 2019; Wang et al., 2019a). The surrounding area comprises farmland
and forests, with no obvious industrial proximity. Two highways are situated about 6 km from the sampling site.
Daily $PM_{2.5}$ samples were collected using a high-volume sampler (model TE-6070 Tisch Inc., Village of Cleves,
OH, USA) at a flow rate of 1.13 $m^3$ $min^{-1}$. The aerosol samples were collected on quartz fiber filters (20.3 cm ×
25.4 cm, Whatman QM/A, Clifton, NJ, USA) that had been pre-heated to 780 °C for 3 h for removing
carbonaceous materials. The sampling equipment was positioned approximately 10 m above the ground level on
a building's rooftop. All sampled filters were enveloped in clean aluminum foils and stored at −20 °C in a freezer
until subsequent analysis in the laboratory. To account for background levels, field blank filters were processed
and analyzed as the same method as the PM samples. All data presented was subtracted by field blank values.
**2.2 NOCs analysis**
A total of 64 $PM_{2.5}$ samples were analyzed to determine the target NOCs in this study. Amines and amino acids
were quantified with the derivatization and analytical procedures by the Waters' AccQ-Tag method (Cohen and
Michaud, 1993; Ho et al., 2015; Ho et al., 2019). For sample extraction, a 4.3 $cm^2$ filter was cut into pieces and
subjected to ultrasonic extraction with 5 mL of Milli-Q water (18 MΩ cm) twice in a water bath at 25 °C. Each
extract was then filtered through a 0.45 μm filter and concentrated to 0.5 mL using a rotary evaporator under
vacuum. The resulting extracts were reacted with 6-aminoquinolyl-N-hydroxysuccinimidyl carbamate (AQC) to
produce fluorescent derivatives. The AccQ-Fluor reagent kit (WAT052880, Waters Corporation, Milford, MA,
USA) consists of AQC and AccQ. Tag borate buffer, and AccQ. Tag Eluent A was used for the derivatization
process. The derivatized sample extracts were reconstituted and stored in a desiccator at room temperature before
analysis. In the HPLC analysis, the derivates and calibration standards were injected into the high-performance
liquid chromatography (HPLC, 1200 Series, Agilent Technology, Santa Clara, CA, USA) equipped with a
fluorescence detector. The sample vials were heated at 55 °C for 10 minutes using the oven within the system.
The mixture was separated using a column (3.9 × 150 mm AccQ.Tag Amino Acid Analysis Silica base) bonded
with a 4-μm C-18 reversed-phase column at 37 °C and detected at an absorption wavelength of 395 nm. The
linearity of the calibrations was assessed by the correlation coefficient ($R^2$ > 0.999), and the minimum detection
limits (MDLs) for the target organic nitrogen species ranged from 0.036 to 0.086 nmol $m^{-3}$. To ensure the
reliability of the analysis, one replicate analysis of the ambient sample was conducted for every 10 samples.
Additionally, ambient samples spiked with known amounts of internal and external standards were analyzed to
assess potential interference from the sample matrix.
For alkyl amides, alkyl nitriles, isocyanates, and cyclic NOCs, the extraction procedures were the same as those
used for the FAAs. After extraction, combination, filtration, and concentration, the extracts were mixed with 50



µL of borate buffer to adjust the pH to 9.1. The solutions were then diluted with a water/acetone mixture (3:1, v/v)
to a final volume of 150 mL. To this mixture, 40 mL of dansyl chloride in acetone and 10 mL of an internal
standard were added. The resulting mixture underwent a derivatization reaction, which involved vortex agitation
for 1 minute and subsequent ultrasound irradiation at 35 °C for 15 minutes, following the method described by
Ruiz-Jiménez et al. (2012). The reaction vials were kept in the dark until the analysis. The derivatized products
were introduced into the HPLC system, which was equipped with a 2.1 × 150 mm C18 column (3.5-µm particle
size, Waters Sunfire), and coupled with an ion-trap mass spectrometer (Esquire 3000, Bruker Daltonics). The
linearity of the calibrations for these compounds was evaluated using the correlation coefficient ($R^2 > 0.999$). The
MDL for the target organic nitrogen species ranged from 0.005 to 0.019 nmol m$^{-3}$.
Urea was identified and quantified using a direct injection method on an HPLC system coupled with a
photodiode array detector (DAD) (1200 Series, Agilent Technology). The separation of urea was achieved using
a 4.6 × 150 mm C18 column (4-µm particle size, Cogent Bidentate), and its detection was performed at an
absorption wavelength of 210 nm (Ho et al., 2019). The calibration of the method exhibited a high correlation
coefficient ($R^2 > 0.999$), indicating a strong linear relationship between the concentration of urea and the detector
response. The MDL for urea was determined to be 0.05 ng mL$^{-1}$, denoting the lowest concentration of urea that
could be reliably detected using the analytical method. By employing this direct injection approach, along with
the specific column and detection parameters, accurate identification and quantification of urea in the samples
were achieved. The high linearity of the calibration and low MDL underscore the sensitivity and reliability of the
method for analyzing urea content in the study.
**2.3 Auxiliary measurements**
Organic carbon (OC), elemental carbon (EC), organic markers including polycyclic aromatic hydrocarbons (PAHs)
and levoglucosan, and elemental components of PM$_{2.5}$ including Ca, Ti, V, Mn, Fe, Cu, As, Br, Pb, and Zn were
also determined (Table S1). Further details regarding the chemical analyses, including processes, accuracies,
precisions, and quality assurance/quality control (QA/QC) procedures of auxiliary data, can be found in Text S1
in Supplement Information.
**2.4 Estimation of secondary organic carbon (SOC)**
In this study, an approach called the minimum $R^2$ (MRS) method was utilized to estimate [SOC] concentration
(Wu and Yu, 2016) which is deduced using the following equations:
$$[SOC] = [OC] - [POC] \qquad (1)$$
$$[POC] = [EC] \times (OC/EC)_{primary} \qquad (2)$$
where [OC] and [EC] represent the measured concentrations, [POC] represents the primary organic carbon
concentrations, and (OC/EC)$_{primary}$ denotes an estimate of the primary OC/EC ratio. We calculated a series of
(OC/EC)$_{primary}$ values to achieve the lowest coefficient of determination ($R^2$) between [SOC] and [EC], as shown
in Fig. S2. This minimization of $R^2$ allows the accurate deduction of SOC levels, considering the relationship
between [SOC] and [EC].

**2.5 Source apportionment**
Source apportionment using Positive Matrix Factorization (PMF) with the multilinear engine (ME-2) was
performed by employing the source finder tool SoFi v6.7 (Canonaco et al., 2013). The analysis involved aligning
daily measurements of seven nitrogen organic classes with concurrent measurements of three carbonaceous



materials (EC, POC, and SOC), one water-soluble inorganic ion ($K^+$), and 10 elements (Ca, Ti, V, Mn, Fe, Cu,
As, Br, Pb, and Zn) in the $PM_{2.5}$ fraction. The characteristics of the input species and the correlation matrix of
each species can be found in Table S2 and Fig. S3, respectively, providing statistical information for the analysis.
Details of the PMF and ME-2 analysis can be found in the supplementary (Text S1). Briefly, we first performed
unconstrained PMF with a factor number of 2-12 and examined the factor profile and time series (Fig. S4-7). 7-
factor factors were determined as the optimum solution (Fig. S8 and S9). To reduce the mixing between the factors,
a constrained PMF analysis using the "a value" approach of the ME-2 solver was applied (Canonaco et al., 2013).
The 7-factor with the constrained matrix is shown in Table S2. The constrained run was performed by adding
constraints in the base run resolved factor profiles so that the tracers are only present in the corresponding sources
(Wang et al., 2019b).

**2.6 Potential source contribution function (PSCF)**
The potential source contribution function (PSCF) was used to identify the likely pollution regions that influenced
PMF factors based on back trajectories. PSCF analysis was performed using Zefir (Petit et al., 2017). Each
trajectory includes a range of latitude–longitude coordinates every 1-hr backward in a whole day. The studying
field is from 20 to 30 °N, and 90 to 105 °E, which includes more than 95% of the area covered by all the paths.
The set of trajectory data for each arriving elevation level contained two trajectories per day. More details of the
PSCF analysis can be found in Text S1.

**2.7 Community Multiscale Air Quality**
The Community Multiscale Air Quality (CMAQ) model (Version 5.4) was applied to assess the transport of
aerosols from neighboring countries in southwest China. The CMAQ model was configured with the aero7 aerosol
module and cb6r5 gas-phase mechanism (Murphy et al., 2021). The model adopted a horizontal grid resolution
of 27 km, consisting of 34 vertical layers.
185       To generate the necessary meteorological fields for the CMAQ simulations, the Weather Research and
Forecasting (WRF) model (version 4.4) was utilized. The initial and boundary conditions for WRF were obtained
from the National Centers for Environmental Prediction (NCEP) Final (FNL) dataset, which is a reanalysis dataset.
For the domestic emission inventory, the Multiresolution Emission Inventory for China (MEIC) was employed.
Additionally, the MIX inventory was used to account for emissions from other countries (Li et al., 2017).
190       Two simulation cases were conducted: one considering only domestic emissions (i.e., MEIC), and the other
considering emissions from both domestic and other countries (i.e., MEIC + MIX). By employing the zero-out
method, the differences between these two cases represented the contribution of emissions from other countries
to the $PM_{2.5}$ levels in the study area. The CMAQ simulations were performed from March 9[th] to March 27[th], 2018,
with the first 3 days considered a spin-up period for the model. The simulation period covered the first two weeks
of the campaign, encompassing the period before and during the initial pollution event from March 22[nd] to March
26[th]. CMAQ reproduced the measured $PM_{2.5}$ at GMG reasonably well when considering both MEIC and MIX in
the emission inventory, with a correlation coefficient of r >0.9 between the modeled and measured $PM_{2.5}$ and a
slope of 0.61 (Fig. S11).



**3 Results and discussion**
**3.1 Overview of NOC Concentration**
Figure 1 illustrates the concentration variations of NOCs, carbonaceous aerosols, and meteorological parameters
in Gaomeigu during the campaign. The daily mass concentrations of NOCs range from 714.4 to 3887.1 ng m$^{-3}$,
with an average of 2119.4 ± 875.0 ng m$^{-3}$. This average is approximately 40% greater than the NOCs concentration
observed at a regional site in Xianghe, China (1270 ng m$^{-3}$) (Wang et al., 2022). The NOCs are classified into
major (> 10% contribution) and minor (< 10% contribution) compounds, as detailed in Table 1, with the major
classes including FAAs, amines, and urea. The average concentrations of these major NOCs are 1922.6 ± 790.5
ng m$^{-3}$, dominated by FAAs (58.9%), followed by amines (28.0%), and urea (13.7%). Minor NOC species such
as alkyl amides, alkyl nitriles, isocyanates, and cyclic NOCs have average concentrations of 45.1 ± 18.6 ng m$^{-3}$,
4.68 ± 1.75 ng m$^{-3}$, 10.9 ± 4.73 ng m$^{-3}$, and 136.2 ± 61.6 ng m$^{-3}$, respectively.
As shown in Fig. 1, the campaign is segmented into five periods (EC1-EC5) based on meteorological conditions
and NOC concentration variations. The clean period featured a temperature consistently above 9 °C and an
average OC concentration of 2137.3 ± 296.7 ng m$^{-3}$. Elevated wind speeds during this period (4.4 ± 1.3 m s$^{-1}$)
enhanced atmospheric dispersion relative to other polluted periods. Notably, average NOC concentration
increased during high NOC concentration periods, reaching 1482.6 ± 346.4 ng m$^{-3}$, which is more than triple the
level observed during the clean period (451.8 ± 65.2 ng m$^{-3}$). Delving into the high NOC concentration periods
individually, EP1 shows the highest aggregate concentration of major NOCs, which is 4.3 to 5.0 times greater
than during the clean period. The NOCs/POC ratios were 0.773 (EP1), 0.774 (EP2), 0.674 (EP3), and 0.638 (EP4),
presenting a stark contrast to the clean period's ratio of 0.503. However, the NOCs/SOC ratio remains relatively
stable across the phases. These trends underscore the significant influence of primary sources during elevated
NOC concentration periods. Conversely, during the clean period, the source of NOCs appears more complex,
suggesting a nuanced interplay of primary and secondary sources. A more in-depth discussions on source
apportionment are provided in Section 3.4.
**3.2 Major NOC Classes**
**3.2.1 Free Amino acids (FAAs)**
During the sampling campaign, the average FFA concentration is 1092.9 ± 443.37 ng m$^{-3}$, in a range of 370.2 and
2033.2 ng m$^{-3}$ (Table 1). This level is comparable with FAAs observed in regions such as rural Guangzhou, China
(Song et al., 2017), Arizona, U.S. (Boreson et al., 2004), and Antarctica's MZ Station, U.S. (Barbaro et al.,
2015) but is higher than in urban/suburban and marine regions like Nanchan, China (Zhu et al., 2020), California,
U.S. (Zhang and Anastasio, 2003b), Qingdao, China (Shi et al., 2010), Hong Kong, China (Ho et al., 2019).
Notably, the average FAAs concentration in this study is approximately four times higher than that reported in
Xianghe, China (Wang et al., 2022).
FAAs are classified into protein-type and non-protein-type categories. Table S3 provides an overview of
protein-type and non-protein-type FAAs, with mean concentrations of 989.5 ± 403.54 ng m$^{-3}$ and 103.3 ± 41.76
ng m$^{-3}$, respectively. Protein-type FAAs, including Asp, Ser, Glu, Gly, His, Thr, Ala, Pro, Cys, Tyr, Val, Met,
Lys, Ile, Leu, and Phe, accounts for 90.5% of total FAAs, with Glycine (Gly) being the most prevalent. These
findings are consistent with previous studies that identified Gly as the predominant FAA in Nanchang (Zhu et al.,
2020), Hong Kong (Ho et al., 2019), and Venice (Barbaro et al., 2011). Non-protein-type FAAs such as β-
alanine (β-Ala), γ-aminobutyric acid (γ-Ala), and ornithine (Orn) also contributed, with β-Ala representing 9.5%
of these FAAs.





Figure 2 illustrates a positive correlation between FAAs and $O_x$ ($NO_2 + O_3$), indicating an association with secondary formation processes post-precursor emissions. The average FAA concentration is 900 ng m$^{-3}$ at $O_x$ levels below 70 ppb but rises above 1200 ng m$^{-3}$ when $O_x$ exceeds 85 ppb. Moreover, FAAs correlate strongly with both POC (r = 0.95) and SOC (r = 0.90), indicating that secondary processes likely influence the FAA formation, despite no obvious direct local emission near the sampling site. This suggests contributions from both primary and secondary sources to the FAA levels observed.

Moreover, Gly comprises 31% of total FAAs and shows a similar positive relationship with $O_x$. The Gly concentration increases from 250 ng m$^{-3}$ when the $O_x$ is below 70 ppb to 400 ng m$^{-3}$ when the $O_x$ is above 85 ppb. Its correlations with POC (r = 0.94) and SOC (r = 0.89) reinforce the impact of secondary formation processes, similar to patterns observed in the North China Plain (NCP) region, China (Wang et al., 2022).

**3.2.2 Amines and urea**

The average concentration of amines during the sampling period is 563 ng m$^{-3}$. Aliphatic amines dominate, contributing 90% of the total amine, while aromatic amines constitute less than 1% (Fig. 3). The remaining 9% includes other amine compounds including ethanolamine, galactosamine, 2-amino-1-butanol, and N-methylformamide. During the pollution episodes, aliphatic amine concentrations exceed 600 ng m$^{-3}$, with a maximum of 1000 ng m$^{-3}$. In contrast, during clean periods, these levels declined to ~200 ng m$^{-3}$. The proportions of aliphatic amines during pollution episodes are 90-91%, which decreases to 84% during clean periods, with an increase in other concentrations.

Methylamine (MA) emerges as the predominant aliphatic amine, constituting 62% of the total aliphatic amines. Ethylamine (EA) follows, contributing 28% to the total aliphatic amines. Dimethylamine (DMA), trimethylamine (TMA), and other amine species together account for the remaining 10%. Both MA and EA exhibit negative correlations with ambient temperature (Fig. 3), indicating the potential influence of temperature on gas-to-particle partitioning. Below 12°C, the average MA concentration is around 400 ng m$^{-3}$, which halves to 200 ng m$^{-3}$ as temperature increases above 18°C. Similarly, EA concentration is higher at lower ambient temperatures, around 195 ng m$^{-3}$ below 12°C, decreasing to 100 ng m$^{-3}$ above 18°C. Given their low molecular weight, MA and EA are more prevalent in the gas phase at elevated ambient temperatures, where they also exhibit enhanced atmospheric reactivity with acids, transforming into other compounds.

Both MA and EA show negative correlations with RH, with elevated concentrations at lower RHs (Fig. 3d). This inverse relationship might be counterintuitive, given that higher RH typically promotes the partitioning of low molecular weight amine into the particle phase. However, MA and EA, being atmospheric reactive amines, are involved in in-particle reactions. Under high RH conditions, increased condensation of acids and/or reactive organic compounds occur, which subsequently react with MA and EA, consuming them and thus establishing a negative correlation with RH.

Urea is identified as the third major NOC species, with an average concentration of 266 ng m$^{-3}$ during the campaign. This value is approximately half that reported at a regional site in the NCP (Wang et al., 2022), though the direct comparison is limited due to spatial and temporal differences. The urea level at this elevated site highlights the notable role of agricultural fertilizers as a potential source. Urea can be released into the atmosphere through agricultural activities and biomass burning, and it can also be formed secondarily in the atmosphere through chemical reactions.



### 3.3 Minor NOC Classes

#### 3.3.1 Alkyl amides and nitriles

In this study, the distributions and sources of alkyl amides in a range of $C_6$-$C_{20}$ were determined in Gaomeigu. Figure S10 illustrates the distribution patterns of these species during the campaign, where the concentrations of n-alkyl amides vary from 1.11 to 7.57 ng m$^{-3}$, reflecting diverse emission sources. These amides can originate from anthropogenic activities such as coal combustion and vehicular traffic, as well as biogenic processes. To distinguish between these sources, we use the carbon preference index (CPI) and the oleamide to stearamide ratio (Cheng et al., 2006). The CPI, calculated as the ratio of the sum of odd-numbered $C_7$-$C_{19}$ alkyl amides to even-numbered C6-20 alkyl amides, helps identify the dominant source: a CPI $\leq 1$ indicates anthropogenic dominance, whereas $>1$ suggests biogenic predominance (Abas and Simoneit, 1996). The results show that the CPI of alkyl amides ranges from 0.46 to 0.75, with an average of 0.61 ± 0.05, emphasizing the anthropogenic impact on their concentrations. Notably, the CPI values do not vary between the periods having low and high NOC concentrations, suggesting consistent alkyl amide sources throughout the campaign, potentially influenced by long-range transport and stable meteorological conditions.

Beyond the CPI, the $R_{18}$, which is a ratio of oleamide ($C_{18:1}$) and stearamide ($C_{18:0}$), serves as an indicator for alkyl amide aging (Wang et al., 2022). This ratio provides insights into the precursor composition, oxidation degradation, and transport processes influencing unsaturated amide concentrations (Nielsen et al., 2012). An $R_{18} < 1$ implies the aging of alkyl amides due to long-range transport, whereas $R_{18} > 1$ indicates local biomass-burning emissions (Cheng et al., 2006). According to the results of this study, $R_{18}$ values range from 0.73 to 2.27, suggesting the alternation between local and long-range transport.

During the sampling period, the average concentration of alkyl nitriles is 4.69 ± 1.75 ng m$^{-3}$ in Gaomeigu. As shown in Table S3, hexadecanenitrile ($C_{16}$) is the most prevalent (0.49 ng m$^{-3}$), followed by tetradecanenitrile ($C_{14}$) (0.45 ng m$^{-3}$). The concentrations of the other analyzed alkyl nitriles are below 0.4 ng m$^{-3}$. The results are constant with the higher concentrations observed at the Xianghe site (Wang et al., 2022). Moreover, the CPI values for alkyl nitriles were between 0.605 to 0.848, with an average of 0.702 ± 0.05, which points out the anthropogenic influence on their levels. During high NOC concentration phases, the CPI values remain constant (i.e., EP1: 0.72, EP2: 0.71, EP3: 0.71, and EP4: 0.72), compared to 0.75 during clean periods. This consistency implies that anthropogenic sources predominantly influence alkyl nitrile concentrations regardless of the pollution levels.

Furthermore, it is important to note that alkyl amides and nitriles might form as secondary products during biomass burning through reactions between ammonia ($NH_3$) and FAAs (Simoneit et al., 2003). The link between biomass burning and the generation of these compounds is reinforced by robust correlations with levoglucosan and $K^+$ in Fig. S3 (r > 0.88, p < 0.01), both recognized as markers for biomass burning (Wang et al., 2018; Liu et al., 2021b). These evidences firmly confirm that biomass burning is a key contributor to the occurrence of alkyl amides and nitriles in the region.

#### 3.3.2 Cyclic NOCs and isocyanates

The average mass concentration of cyclic NOCs is 136.2 ng m$^{-3}$. This study identified five cyclic NOCs (Table S3), with caprolactam being the most prevalent at 54.2 ng m$^{-3}$ (39.8% of the total cyclic NOC), which is commonly used in commercial manufacturing processes and lysine synthesis (Cheng et al., 2006). Other cyclic NOCs include isoindole-1,3-dione (50.7 ng m$^{-3}$, 37.2%,), N-butyl-benzen-sulfonamide (NBBS) (22.1 ng m$^{-3}$, 16.2%,), N,N-diethyl-m-toluamide (DEET) (5.79 ng m$^{-3}$, 4.3%,), and benzothiazolone (3.36 ng m$^{-3}$, 2.5%). These compounds are known to pose health risks (Cheng et al., 2006; Balducci et al., 2012), which primarily originate from





industrial and agricultural activities(Wang et al., 2022; Richardson and Ternes, 2018; Trapp and Eggen, 2013). In
comparison with the findings of the Xianghe site (Wang et al., 2022), the concentrations of cyclic NOCs in this
study are lower, indicating the lower contributions of industrial sources. During the four high NOC emission
periods, the concentrations of cyclic NOCs are 2-4 times higher than those during the clean period, suggesting the
influence of pollution levels.
Isocyanates, commonly used in polyurethane resin production, are associated with several health threats,
including asthma, allergies, and skin reactions (Lesage et al., 2001). The average total mass concentration of
eight isocyanates is $10.89 \pm 4.73$ ng m$^{-3}$ (Table 1) while the individual concentration of each isocyanate is given
in Table S3, including methyl isocyanate (MIC), toluene-2,4-diisocyanate (2,4-TDI), toluene-2,6-diisocyanate
(2,6-TDI), isophorone diisocyanate (IPDI), 1,6-hexamethylene diisocyanate (1,6-HDI), ethyl isocyanate (EIC),
phenyl isocyanate (PHI), and propyl isocyanate (PIC). Among these, TDI and HDI are predominantly used in
industry (Hejna et al., 2024). TDI is commonly utilized in various foam products (Akindoyo et al., 2016), while
HDI is essential in polyurethane paints and coatings (Golling et al., 2019). The presence of these isocyanates in
numerous products is linked to heightened health hazards, such as skin allergies, atopic dermatitis, and various
respiratory diseases (Nawrot et al., 2008).
**3.4 Sources apportionment of NOCs**
In this study, a constrained PMF analysis was applied to identify the sources of NOCs, which include biomass
burning, coal combustion, industry-related sources, crustal sources, traffic emissions, agriculture activities, and
secondary sources (Fig. 4).
Factor 1, attributed to biomass burning, was characterized by high loadings of $K^+$ (84.3%) and levoglucosan
100%), recognized tracers for biomass-burning activities (Liu et al., 2021a; Lin et al., 2018). This factor also has
a notable Zn content (38.7%), indicative of wood burning (Salam et al., 2013). Biomass burning contributes 26.3%
to the total NOCs, emerging as the second-largest emission category. Factor 2, associated with coal combustion,
exhibits substantial loadings of As and also contains Cu, Pb, and EC. As and Pb are typical tracers of coal
combustion (Qin et al., 2019), and Cu is also associated with coal combustion (Hsu et al., 2016). Factor 3 is
recognized as industry-related emissions which is characterized by high loading of cyclic NOCs and isocyanates,
which are synthetic compounds (Wang et al., 2022). It also exhibits a significant characteristic value of Pb, which
can be released during industrial processes (Wang et al., 2015). This factor accounts for 7.6% of NOCs. Factor
4, characterized by crustal sources, had high loadings of Ti and moderate loadings of Mn, Fe, Ca, and arabitol.
These elements are acknowledged as crustal constituents (Gosselin et al., 2016), and arabitol is typically released
from soil fungal spores (Wang et al., 2018), contributing to 6.1% of the total NOCs. Factor 5, linked to traffic
emissions, showed high loadings of V, Br, Zn, and Cu. V acts as an indicator for heavy oil combustion in marine
vessels (Bian et al., 2018), and Br is a tracer of motor vehicle emissions (Guo et al., 2009). Emissions of Zn and
Cu are associated with brake, tire, and road wear (Salameh et al., 2018; Liu et al., 2021a). Factor 6, named
agriculture activities, exhibited relatively high loading of urea and moderate loadings of $K^+$, Ca, and Mn in NOCs.
These elements are commonly used in agriculture (Ge et al., 2011), with $K^+$ being crucial for plant growth and
metabolic functions (Meena et al., 2014), and Mn playing roles in plant oxidation-reduction (Gonçalves et al.,
2022). This factor accounted for a portion of NOCs. Factor 7, ascribed to secondary sources, demonstrated
considerable influence on the SOC (Secondary Organic Carbon) variation. It was responsible for 30.2% of the
NOCs, emerging as the predominant emission source, highlighting the role of secondary production in both local
and regional pollutant formation.
Figure 5 illustrates the average contributions of the seven identified sources to each NOC species and the total
NOC. The analysis demonstrated that secondary sources and biomass burning were predominant, together



constituting over 50% of total NOCs (Figure 5a). Specifically, for FAAs (Figure 5b), secondary sources (39.6%)
and biomass burning (37.3%) are the two major contributors, while other sources accounted for less than 10%.
The notable influence of secondary sources and biomass burning on FAAs could be attributed to increased
transportation and biomass/wildfire heating in the region, consistent with findings in a previous study (Zhang et
al., 2018).
In the context of amines, agriculture activities make a notable contribution (18.8%), twice as high as its
contribution to FAAs (9.3%). For alkyl amides and nitriles, secondary sources and biomass burning were the
primary contributors, each surpassing 30%. This contrasts with findings from another study in a different Chinese
region where biomass burning is predominant in these NOC categories (Wang et al., 2022). These significant
contributions from biomass burning and secondary sources underscore the impact of regional transportation on
NOC sourcing within this area.

### 3.5 Influence from long-range transport and biomass burning in Gaomeigu

Figure 6 presents the spatial distribution of $PM_{2.5}$ concentrations during the high NOC events, analyzing
two scenarios: one with only domestic emissions (MEIC-China) and another incorporating both domestic
and foreign emissions (MEIC-China + MIX). With solely domestic emissions considered, $PM_{2.5}$ levels at the
GMG and across the broader Tibet region, as well as western Sichuan and Yunnan, were relatively low, not
exceeding 5 µg m$^{-3}$ (Fig. 6a). However, incorporating international transport into the analysis revealed a
significant increase in $PM_{2.5}$ levels at GMG, where daily concentrations exceeded 20 µg m$^{-3}$ (Fig. 6c).
Similarly, elevated $PM_{2.5}$ concentrations, reaching above 40 µg m$^{-3}$, were observed in southeast Tibet and
western Sichuan and Yunnan. Figure 6b presents the relative contributions of domestic and international
emissions at GMG. The contribution from international transport varied from 25% to 92%, overshadowing
domestic sources, which did not exceed 25% for most of the time. Notably, during the high NOC event, the
contribution from international transport increased to over 80% for the study area (Fig. 6d).
The emission inventory used in this study did not include data on NOCs; hence, NOCs were not explicitly
simulated in the CMAQ model. However, the marked influence of international transport indicates that
$PM_{2.5}$-bound NOC species likely originated from international sources, corroborated by PSCF analysis
linking NOCs to specific PMF factors (Fig. S12). The contribution hotspots in India and Myanmar indicate
that the long-range transport of biomass-burning emissions to the study area is facilitated by prevailing winds.
Conversely, secondary NOC sources were predominantly linked to air masses from Myanmar, implying
proximate secondary formation through atmospheric reactions of precursor gases and pollutants. The
complex atmospheric chemistry leading to secondary NOCs includes the oxidation of precursor compounds
such as volatile organic compounds (VOCs) and nitrogen oxides ($NO_x$).
Similar spatial patterns were observed for factors related to coal combustion, industry-related sources,
crustal sources, traffic emissions, and agricultural activities. This implies that their contributions were
associated with the proximity of the sampling site to their respective source origins. For instance, NOCs
related to coal combustion were potentially transported from the nearby mining or industrial areas, while
industry-related sources could have originated from regional transmission or industrial activities in the
vicinity. Crustal sources, which involve the resuspension of dust particles, could be influenced by local soil
conditions and wind patterns.



**4 Conclusions**

In conclusion, this study provides valuable insights into the composition, sources, and transport of NOCs in the study area. The average daily mass concentrations of NOCs during the campaign ranged from 714.4 to 3887.1 ng m$^{-3}$, with an average of 2119.4 ± 875.0 ng m$^{-3}$. The major NOC species include free amino acids (FAAs), amines, and urea, accounting for 58.9%, 28.0%, and 13.7% of the major NOCs, respectively. Minor NOC species such as alkyl amides, alkyl nitriles, isocyanates, and cyclic NOCs were also identified. The PMF analysis revealed seven distinct sources of PM$_{2.5}$, with biomass burning and secondary sources as the primary contributors to total NOCs. Biomass burning sources exhibited hotspots of contribution from India and Myanmar, indicating long-range transport. Secondary sources, predominantly originating from Myanmar, suggested the formation of NOCs during the transport. This is confirmed by the CMAQ modeling. The study also revealed the possible aging of NOCs from biomass-burning sources as they approached the measurement site, highlighting the impact of atmospheric transformation processes. Contributions from industry-related sources, crustal sources, and agricultural activities were influenced by both regional transmission and local emissions in the vicinity of the sampling site. Overall, this research highlights the complex nature of NOCs and their sources, emphasizing the interplay between long-range transport, regional emissions, atmospheric chemistry, and local influences. These findings contribute to our understanding of air pollution dynamics and provide a basis for developing targeted mitigation strategies and policies to reduce NOC emissions and their impacts on air quality and human health in the study area and similar regions.

**Declaration of competing interest**

The authors declare that they have no known competing financial interests or personal relationships that could have appeared to influence the work reported in this paper.

**Credit authorship contribution statement**

Meng Wang: Conceptualization, Methodology, Validation, Formal Analysis, Writing - Original Draft.
Qiyuan Wang: Conceptualization, Writing - Review and Editing, Funding Acquisition.
Steven Sai Hang Ho: Formal analysis, Writing - Review, and Editing.
Jie Tian: Investigation.
Yong Zhang: Investigation, Formal analysis.
Shun-cheng Lee: Resources.
Junji Cao: Conceptualization, Writing - Review and Editing, Funding Acquisition, Supervision.

**Acknowledgments**

This work was supported by the Second Tibetan Plateau Scientific Expedition and Research Program (STEP) (2019QZKK0602), the National Natural Science Foundation of China (42305122), the Strategic Priority Research Program of the Chinese Academy of Sciences (XDB40000000), and the Natural Science Basic Research Program of Shaanxi (2023-JC-JQ-23). Qiyuan Wang also acknowledged the support from the Youth Innovation Promotion



Association of the Chinese Academy of Sciences.

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



**Table 1** Concentration levels of chemical compounds and groups in Gaomeigu, China. (ng m$^{-3}$).

| Species | Mean | SD[a] | Min[b] | Max[c] |
|---|---|---|---|---|
| **NOCs (ng m$^{-3}$)** | | | | |
| **Major Compound Classes** | | | | |
| **FAAs** | | | | |
| Protein FAAs | 989.5 | 403.5 | 337.8 | 1857.5 |
| Non-protein FAAs | 103.3 | 41.8 | 32.5 | 206.8 |
| **Total FAAs** | 1092.9 | 443.4 | 370.2 | 2033.2 |
| **Amines** | | | | |
| Aliphatic Amines | 508.9 | 225.9 | 158.5 | 1032.2 |
| Aromatic Amines | 2.6 | 1.0 | 1.0 | 5.3 |
| Other Amines | 51.8 | 16.8 | 29.2 | 100.7 |
| **Total Amines** | 563.3 | 240.2 | 190.2 | 1113.5 |
| **Urea** | 266.4 | 119.0 | 79.4 | 588.8 |
| **Total Major Compound** | 1922.6 | 790.5 | 649.0 | 3543.7 |
| | | | | |
| **Minor Compound Classes** | | | | |
| **Amides** | | | | |
| Alkyl amides (Odd) | 13.1 | 5.8 | 4.1 | 26.6 |
| Alkyl amides (Even) | 21.4 | 8.9 | 6.6 | 41.2 |
| **Total Alkyl amides** | 45.1 | 18.6 | 14.9 | 84.6 |
| | | | | |
| **Nitriles** | | | | |
| Alkyl nitriles (Odd) | 1.9 | 0.7 | 0.8 | 3.5 |
| Alkyl nitriles (Even) | 2.7 | 1.0 | 1.0 | 4.8 |
| **Total Alkyl nitriles** | 4.7 | 1.7 | 1.8 | 8.2 |
| | | | | |
| **Cyclic NOCs** | 136.2 | 61.6 | 42.1 | 291.9 |
| **Isocyanates** | 10.9 | 4.7 | 3.3 | 23.2 |
| **Total Minor Compound** | 196.8 | 86.1 | 65.4 | 404.4 |
| **Total NOCs** | **2119.4** | **875.0** | **714.4** | **3887.1** |

[a]SD represents standard deviation. [b]Min and [c]Max donate "minimum and maximum, respectively.



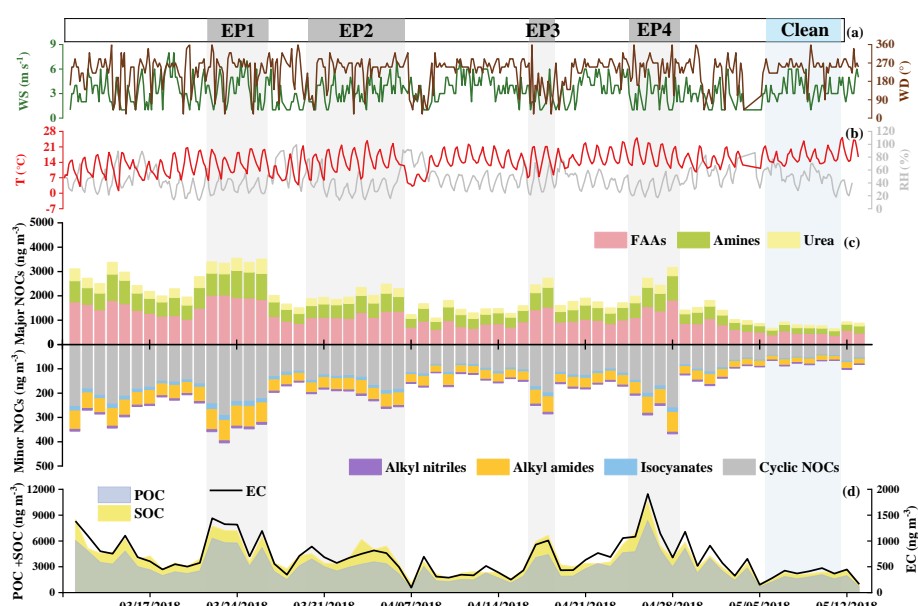


**Figure 1** Hourly variations in meteorological parameters and daily chemical compositions of NOCs during different events in Gaomeigu in 2018 (EP1: 3/22 to 3/26; EP2: 3/30 to 4/6; EP3: 4/17 to 4/18; EP4: 4/25 to 4/28; Clean period: 5/6 to -5/11).





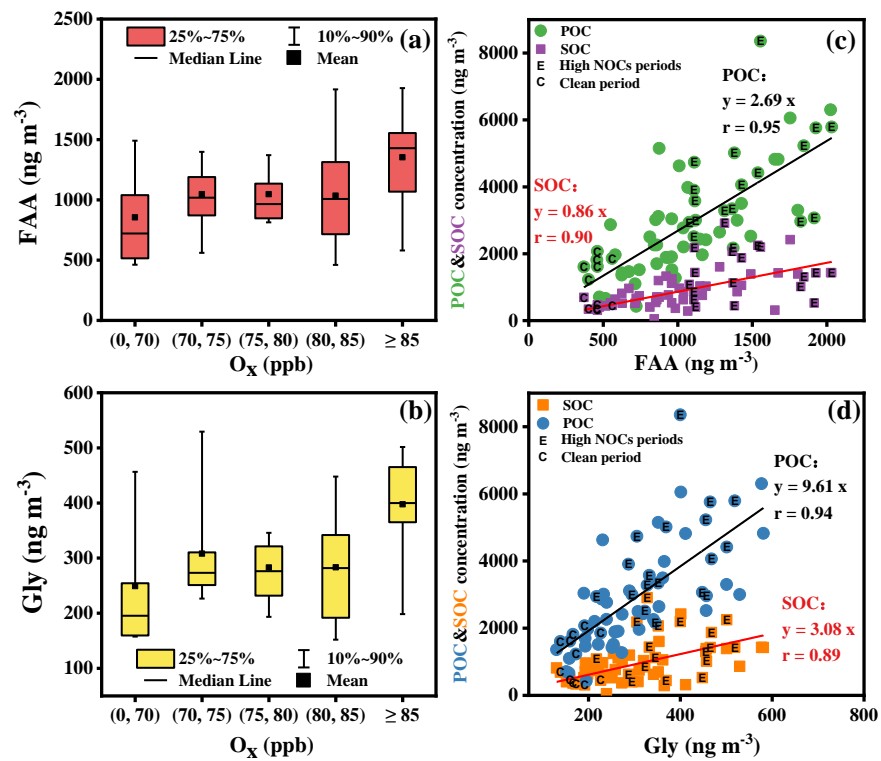

**Figure 2** (a) FAA dependence of $O_3$; (b) Gly dependence of $O_3$; Correlation plots of POC&SOC concentration versus (c) FAA, and (d) Gly. The box represents the 25th (bottom) and 75th percentiles (top), and the box-whisker data represent the range from 10th to 90th percentiles.




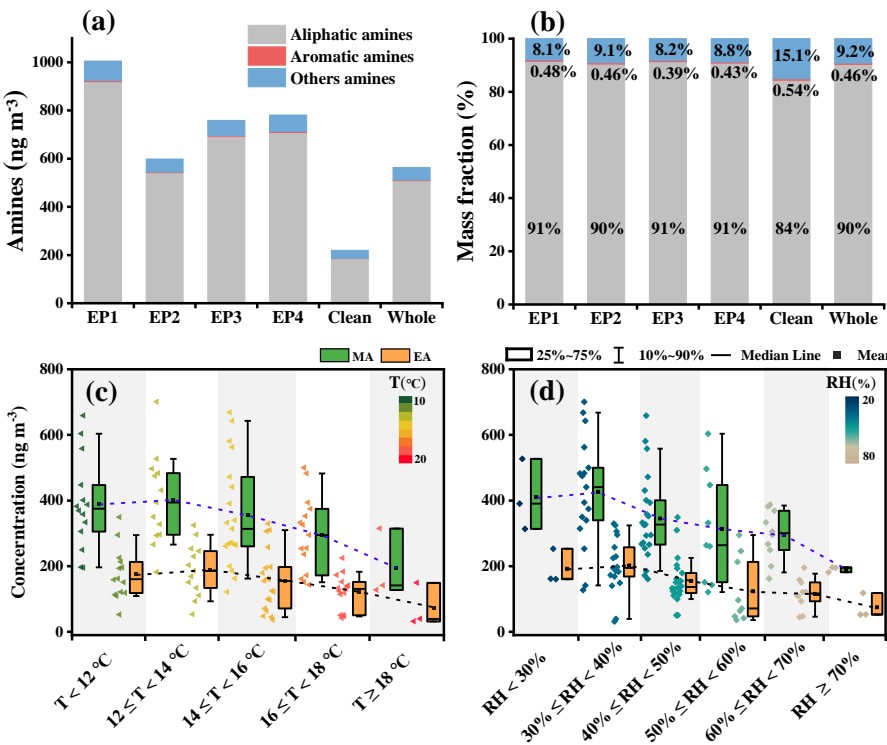

**Figure 3** (a) Concentration and (b) composition of amines. (c) Temperature dependence of EA and MA, and (d) RH dependence of EA and MA. The box represents the 25th (bottom) and 75th percentiles (top), and the box-whisker data represents the 10th to 90th percentiles.



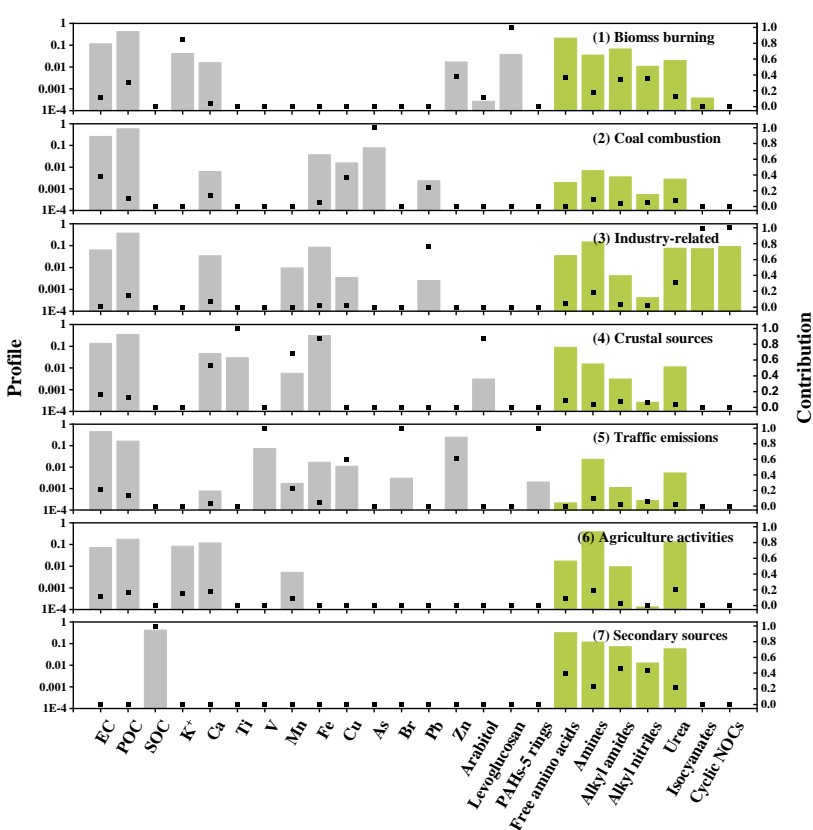


**Figure 4** The factor profiles and explained variations in the ME-2 modeling.





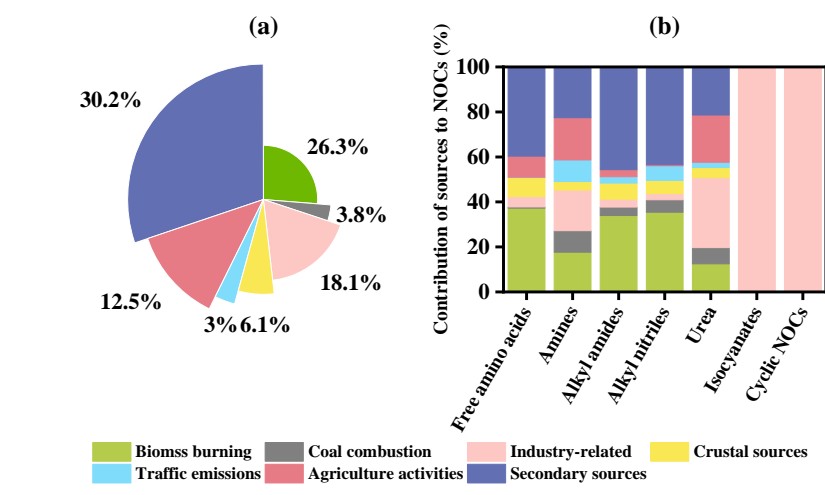


**Figure 5** Contributions of each source to (a) total NOCs; and (b) seven classes NOC species.





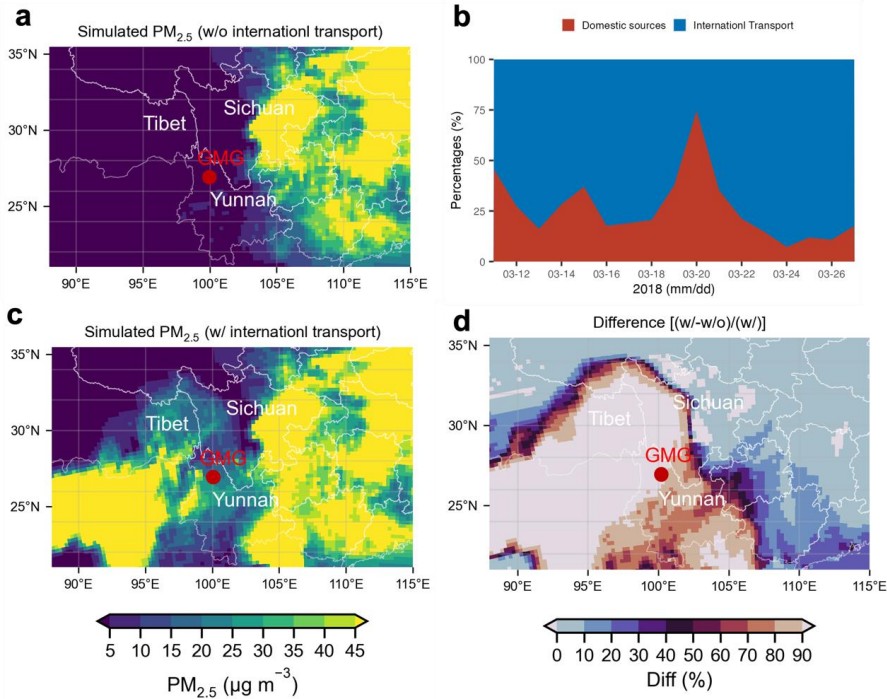


**Figure 6.** (a) Distribution of PM$_{2.5}$ concentrations resulting solely from China's domestic emissions (MEIC-China only); (b) Proportionate contributions of domestic versus international PM$_{2.5}$ transport during the simulation window of March 11$^{th}$ -27$^{th}$ 2018; (c) Distribution of PM$_{2.5}$ incorporating both domestic and international transport influences (MEIC-China+MIX); (d) Difference of contribution of international transport to PM$_{2.5}$ concentrations, derived from the differential analysis [(c)-(a)]/(c).

677