# Peer review of "Dominant Influence of Biomass Combustion and Cross-Border Transport on Nitrogen-Containing Organic"

_EGUsphere, 2024_

## Author Comment (AC1)

The authors thank the editor and anonymous referees for reviewing our manuscript, and particularly providing valuable comments and suggestions. Our responses in form of point-by-point are given.

The authors reported measurement results of nitrogen-containing organic compounds (NOCs) in PM$_{2.5}$ at a regional background site in South Tibetan Plateau. Careful speciation and quantification on NOCs with 60+ samples over 2 months of time span. Concentration levels and variations of main compound classes, including free amino acids, amines and urea, as well as relatively minor ones (in terms of mass concentrations) such as alkyl amides, nitriles, cyclic NOCs, and isocyanates, were discussed. In addition, concentrations of these NOCs and those of other particulate pollutants (e.g, metals and EC/OC) were used to apportion the sources of NOCs at this site. Results suggested that biomass burning and secondary formation were main contributors of NOCs at this site. Furthermore, regional model was used to evaluate the contribution of cross-boundary transport to particulate matter at this site, hinting that such contribution is also important for the NOCs measured. The study is of importance to understand the climate-related pollutants in the less-explored region of Tibetan Plateau. The chemical analysis and data interpretation in this work in rigorous. The manuscript is also clear to follow. I have a few comments as below.

Response: We thank the referee for the positive comments.

Can the authors briefly justify why nitro-aromatic compounds were not included in this study? They are also light-absorbing and can affect climate. The authors hinted on secondary formation of this class of compounds in P11/L394. Is it that the NOx and aromatic VOC level at the regional background site are not high enough to make them important at this site?

Response: We appreciate the reviewer's insightful comment regarding the exclusion of nitro-aromatic compounds in our study. Nitro-aromatic compounds are indeed recognized for their light-absorbing properties and potential impacts on climate. Their secondary formation in the atmosphere, particularly through photochemical reactions involving NOx and aromatic volatile organic compounds (VOCs), is well-documented. Below are the two reasons why they were not included in this study.

1. The methodologies and analytical techniques employed in our study were optimized for the detection and quantification of specific NOCs such as amines, amino acids, and urea. Including a broader range of compounds such as nitro-aromatic compounds would have required additional specific analytical protocols, possibly complicating the study without substantially enhancing the core findings related to the primary sources and impacts of NOCs in the TP region.

2. The regional background site at Gaomeigu, situated at a remote high-altitude location, has relatively low levels of NOx and aromatic VOCs (Wu et al., 2023; Zhu et al., 2020) compared to urban or industrial areas. The low ambient concentrations of these precursors reduce the likelihood of significant secondary formation of nitro-aromatic compounds in this specific environment. Consequently, the concentrations of nitro-aromatic compounds were not expected to be a major factor influencing the overall

NOC levels and their climatic impacts in this region.

Despite the exclusion of nitro-aromatic compounds, we acknowledge their importance and the potential value of including them in future studies.

In the revised text in Section 3.5, it now reads, "…The complex atmospheric chemistry leading to secondary NOCs includes the oxidation of precursor compounds such as volatile organic compounds (VOCs) and nitrogen oxides ($NO_x$). Other NOCs that were not measured in this study, such as nitro-aromatics, were likely contributing to the NOCs and will be the focus of future research..."

I have some reservation on using regional model to estimate $PM_{2.5}$, and then infer that NOCs are also dominantly transported from nearby regions. High $PM_{2.5}$ might not necessarily mean high NOCs. It is better to build a stronger linkage between the model results of $PM_{2.5}$ and the PSCF results of NOCs, such that the conclusion of regional transport of NOCs would be more convincing.

Response: We appreciate the reviewer's critical assessment regarding the use of a regional model to estimate $PM_{2.5}$ and the inference that NOCs are dominantly transported from nearby regions.

While our regional air quality model primarily estimates $PM_{2.5}$ concentrations, we performed additional analyses to correlate these model results with measured NOC concentrations. As shown in the revised Fig. S12, during pollution periods, NOCs levels were also higher with biomass burning and secondary sources contributing over half of the total NOCs. The PSCF analysis identified source regions contributing to high NOC concentrations, which coincided with regions indicated by the $PM_{2.5}$ transport model, showing similar patterns of PSCF (Fig. S14). This convergence of evidence supports the hypothesis that regional transport mechanisms play a significant role in NOC distribution in the TP.

In the revised text in Sect. 3.5, it now reads, "…during the high NOC events, during high NOC events, such as in EP1, where biomass burning and secondary sources contributed over half of the total NOCs (Fig. S12), the contribution from international transport increased to over 80% for the study area (Fig. 6d)…"

Also, "...However, the marked influence of international transport indicates that $PM_{2.5}$-bound NOC species likely originated from international sources, corroborated by PSCF analysis linking NOCs to specific PMF factors (Fig. S13), and by the observed correlation between bulk PM2.5 and total NOCs (Fig. S14)..."

[Figure]

**Fig. S12** Time series of the PMF factors and their contribution during episode periods (EP1-EP4) and clean period in Gaomeigu. The time series of PM$_{2.5}$ concentrations is also shown on the right y-axis.

[Figure]

**Fig. S14** PSCF patterns of PM2.5, OC and NOC, highlighting similar hotspots from international transport.

I am also a bit confused about how free amino acids are formed secondarily. Do you mean the processes of breaking down proteins into free amino acids, or converting, say amines, into amino acids by introducing the COOH group via oxidation? Please clarify.
Response: The secondary formation of free amino acids in aerosols refers primarily to the processes of breaking down proteins into free amino acids. This can occur through several mechanisms, including direct photolysis, photochemical hydrolysis, and enzyme-based hydrolysis. These processes have been documented in previous studies (Mopper and Zika, 1987; Milne and Zika, 1993; Song et al., 2017). Given that the sampling site is subject to long-range transport, it is likely that free amino acids were secondarily produced by the breakdown of proteins from these processes.
In the revised text in Section 3.2, it now reads, "…Secondary formation of FAAs from proteins can occur through several mechanism, including direct photolysis, photochemical hydrolysis, and enzyme-based hydrolysis (Mopper and Zika, 1987; Milne and Zika, 1993; Song et al., 2017). Given that the sampling site is subject to long-range transport (discussed in Sect. 3.5), it is likely that free amino acids were secondarily produced by the breakdown of proteins during the transport..."

References:

Milne, P. J. and Zika, R. G.: Amino acid nitrogen in atmospheric aerosols: Occurrence, sources and photochemical modification, Journal of Atmospheric Chemistry, 16, 361-398, 10.1007/BF01032631, 1993.

Mopper, K. and Zika, R. G.: Free amino acids in marine rains: evidence for oxidation and potential role in nitrogen cycling, Nature, 325, 246-249, 10.1038/325246a0, 1987.

Song, T., Wang, S., Zhang, Y., Song, J., Liu, F., Fu, P., Shiraiwa, M., Xie, Z., Yue, D., Zhong, L., Zheng, J., and Lai, S.: Proteins and Amino Acids in Fine Particulate Matter in Rural Guangzhou, Southern China: Seasonal Cycles, Sources, and Atmospheric Processes, Environ. Sci. Technol., 51, 6773-6781, 10.1021/acs.est.7b00987, 2017.

Wu, X., Sun, W., Huai, B., Wang, L., Han, C., Wang, Y., and Mi, W.: Seasonal variation and sources of atmospheric polycyclic aromatic hydrocarbons in a background site on the Tibetan Plateau, Journal of Environmental Sciences, 125, 524-532, https://doi.org/10.1016/j.jes.2022.02.042, 2023.

Zhu, C.-S., Li, L.-J., Huang, H., Dai, W.-T., Lei, Y.-L., Qu, Y., Huang, R.-J., Wang, Q.-Y., Shen, Z.-X., and Cao, J.-J.: n-Alkanes and PAHs in the Southeastern Tibetan Plateau: Characteristics and Correlations With Brown Carbon Light Absorption, J. Geophys. Res. Atmos., 125, e2020JD032666, https://doi.org/10.1029/2020JD032666, 2020.

P4/L120: define FAAs here.

Response: Now defined. It reads, "...the free amino acids (FAAs)..."

P5/L135: why not using nmol/m^-3 that is consistent with those in the previous paragraphs?

Response: The units used (nmol m$^{-3}$) in the experimental section are the conventional expressions employed in purely chemical analytical methods to represent the concentration of a solution. In subsequent sections, we use units (ng m$^{-3}$) that more directly indicate the mass concentration of NOCs in the air. This approach facilitates comparison with other chemical components of atmospheric particulate matter by using a consistent unit standard.

P6/168: add "solution" after "7-factor".

Response: Added, it now reads "...The 7-factor solution with the constrained matrix is shown in Table S2..."

P7/L210: "EC" or "EP"?

Response: Revised, it should be "EP", it now reads "As shown in Fig. 1, the campaign is segmented into five periods (EP1-EP5) based on meteorological conditions and NOC concentration variations."

P8/L276-278: citation needed.

Response: We add some references about the source of urea in Sec.3.2.2, Line 276-278, it now reads, "Urea can be released into the atmosphere through agricultural activities and biomass burning (Wang et al., 2022), and it can also be formed secondarily in the atmosphere through chemical reactions (Leung et al., 2024)."

Wang, M., Wang, Q., Ho, S. S. H., Li, H., Zhang, R., Ran, W., Qu, L., Lee, S.-c., and Cao, J.: Chemical characteristics and sources of nitrogen-containing organic compounds at a regional site in the North China Plain during the transition period of autumn and winter, Science of The Total Environment, 812, 151451, https://doi.org/10.1016/j.scitotenv.2021.151451, 2022.

Leung, C. W., Wang, X., and Hu, D.: Characteristics and source apportionment of water-soluble organic nitrogen (WSON) in PM2. 5 in Hong Kong: with focus on amines, urea, and nitroaromatic compounds, Journal of Hazardous Materials, 133899, 2024.

P9/L287: subscript for 6 and 20 to be consistent with the notation earlier in the sentence.

Response: Revised, it now reads, "The CPI, calculated as the ratio of the sum of odd-numbered $C_7$-$C_{19}$ alkyl amides to even-numbered $C_6$-$C_{20}$ alkyl amides, helps identify the dominant source".

P9/L297: provide mean +/- standard deviation as in the previous paragraph?

Response: We added the mean +/- standard deviation of $R_{18}$ in Page 9, Line 298. It now reads $R_{18}$ values range from 0.73 to 2.27, with an average of $1.36 \pm 0.35$, suggesting the alternation between local and long-range transport."

P9/L311: remove "firmly".

Response: removed.

P9/L319: replace "which" with "and they".

Response: Revised, it now reads "These compounds are known to pose health risks (Cheng et al., 2006; Balducci et al., 2012), and they primarily originate from industrial and agricultural activities (Wang et al., 2022; Richardson and Ternes, 2018; Trapp and Eggen, 2013)".

P10/L357: how much is "a portion"?

Response: We added the data to describe the proportion of factor 7 within the total NOCs in Page 10, Line 357. It now reads "This factor accounted for approximately 13% of NOCs".

---

## Author Comment (AC2)

The authors thank the editor and anonymous referees for reviewing our manuscript, and particularly providing valuable comments and suggestions. Our responses in form of point-by-point are given.

The manuscript provides a comprehensive analysis of nitrogen-containing organic compounds (NOCs) in fine particulate matter (PM$_{2.5}$) in the southeastern Tibetan Plateau (TP). The authors have conducted a detailed field study and employed robust analytical methods to identify the sources and concentrations of NOCs, emphasizing the significant impact of biomass combustion and cross-border transport. The study is well-structured, and the results are important for understanding the atmospheric chemistry and climate implications in this sensitive high-altitude region. I recommend the manuscript for publication after addressing these minor points.

Response: We thank the referee for the positive comment.

Abstract: the abstract would benefit from a brief mention of the specific analytical techniques used to quantify NOCs, which would provide readers with a better understanding of the study's methodological robustness.

Response: We agree that including a brief mention of the specific analytical techniques used to quantify NOCs in the abstract would enhance the readers' understanding of the study's methodological robustness. We will revise the abstract to include this information. It now reads, "... We conducted field sampling at a regional background sampling site in Gaomeigu, in the southeastern margin of TP from March 11$^{th}$ to May 13$^{th}$ in 2017, followed by laboratory analysis of the NOCs collected on the filters..."

Introduction: The introduction provides a good background on the significance of NOCs and the TP region. It would be helpful to include a brief discussion on the potential implications of NOCs on local human health and ecosystems, in addition to their climatic impact.

Response: We appreciate your suggestion to include a discussion on the potential implications of NOCs on local human health and ecosystems. This addition will provide a more comprehensive overview of the significance of NOCs beyond their climatic impact.

In the revised Introduction, it now reads, "...The increased input of reactive nitrogen from human activities, such as fertilizer production, adversely affects terrestrial and aquatic ecosystems and human health by impacting air, soil, and water quality (De Vries, 2021). These effects have profound implications for atmospheric chemistry and climate, necessitating a deeper understanding of NOC sources and atmospheric processes in the climate-sensitive region of TP..."

line 232: provide a more detailed explanation of the criteria used to segment the campaign into EC1-EC5 periods. For instance, specifying the exact meteorological parameters and concentration thresholds that define each period would enhance clarity.

Response: The criteria for segmenting the campaign into EP1-EP5 periods were based

primarily on mass concentration thresholds. Specifically, high pollution episodes were identified by NOC concentrations that were 4-5 times higher than those observed during clean periods. Meteorological parameters and their impacts on regional transport and secondary formation processes are discussed in detail later in the manuscript.

line 254-261: The mean concentrations of protein-type and non-protein-type FAAs are provided, but it would be useful to discuss the potential reasons for the observed differences in their concentrations. For example, what environmental or biological processes might account for the higher prevalence of protein-type FAAs.

Response: We appreciate your suggestion to discuss the potential reasons for the observed differences in the concentrations of protein-type and non-protein-type FAAs. In our study, the higher prevalence of protein-type FAAs could be attributed to several environmental and biological processes including source contribution, atmospheric processes, and meteorological conditions. These aspects were discussed in Section 3.4 and 3.5.

line 298: add references for "…Urea can be released into the atmosphere through agricultural activities and biomass burning, and it can also be formed secondarily in the atmosphere through chemical reactions…"

Response: Now added.

line 329: While Simoneit et al. (2003) is cited for the formation mechanisms, it would be useful to reference additional studies that have observed similar formations of alkyl amides and nitriles in biomass burning contexts. This would help to further validate the findings and place them within a broader research context.

Response: We have cited the following papers.

Munila Abudumutailifu, Xiaona Shang, Lina Wang, Miaomiao Zhang, Huihui Kang, Yunqian Chen, Ling Li, Ruiting Ju, Bo Li, Huiling Ouyang, Xu Tang, Chunlin Li, Lin Wang, Xinke Wang, Christian George, Yinon Rudich, Renhe Zhang, Jianmin Chen. Unveiling the Molecular Characteristics, Origins, and Formation Mechanism of Reduced Nitrogen Organic Compounds in the Urban Atmosphere of Shanghai Using a Versatile Aerosol Concentration Enrichment System. *Environmental Science & Technology* **2024**, *58* (16) , 7099-7112.

Ma, Y. J., Xu, Y., Yang, T., Xiao, H. W., and Xiao, H. Y.: Measurement report: Characteristics of nitrogen-containing organics in PM2.5 in Ürümqi, northwestern China – differential impacts of combustion of fresh and aged biomass materials, Atmos. Chem. Phys., 24, 4331-4346, 10.5194/acp-24-4331-2024, 2024.

line 383: While other sources accounted for less than 10%, it would be beneficial to briefly mention what these sources are and their potential impact. Even minor contributors can provide important context for a comprehensive understanding of NOC

sources.

Response: We have now mentioned these sources. It now reads, "... Specifically, for FAAs (Figure 5b), secondary sources (39.6%) and biomass burning (37.3%) are the two major contributors, while other sources accounted for less than 10% including agriculture activities, crustal sources, industry-related, coal combustion, and traffic emissions..."

Conclusion: Suggest areas for future research that could build on this study. For instance, further investigation into the specific chemical pathways of NOC formation during transport, or more detailed source apportionment studies in different regions, could be valuable.

Response: We agree that identifying areas for future research would enhance the conclusion and provide direction for subsequent studies.

In the revised conclusion section, it now reads, "... For future research, we suggest further investigation into the specific chemical pathways involved in the formation of NOCs during atmospheric transport, which could involve controlled laboratory experiments and field studies. Additionally, more detailed source apportionment studies in different regions, including urban, rural, and remote areas, would provide a comprehensive understanding of the sources and contributions of NOCs. By addressing these areas, future research can further enhance our understanding of NOCs and inform effective policy measures to mitigate their adverse effects..."

Figure 1: There is a minor typographical error in the description of the clean period: "5/6 to -5/11" should be corrected to "5/6 to 5/11". Consistency in date formats will prevent confusion.

Response: Revised.

---

## Author Response (AR2)

Thank you for your revised manuscript. You have addressed the questions and comments of the referees and I agree that this work will make a valuable contribution to the community. However, before I can accept it for publication in ACP, I have an additional comment regarding the exclusion of nitro-aromatic compounds in your study.

The authors thank the editor for reviewing our manuscript, and particularly providing valuable comments and suggestions. Our responses in form of point-by-point are given.

I agree with the authors that the large diversity of nitrogen-containing components in the particle phase makes it difficult to encompass all those species in one study. A selection of representative species could still support your key findings on the dominant influence of biomass combustion and cross-border transport on these species. However, it is important to note that nitro-aromatic compounds are not only oxidation products of aromatics like toluene. They are also emitted directly during biomass burning and can be produced from the oxidation of biomass burning emissions like phenolic compounds. Given the significant influence of biomass combustion at your sampling site, one would expect nitro-aromatics to be a considerable fraction of the particles.

Response: We appreciate your feedback and agree with your comments. In our future studies, we will focus on NOCs, including nitro-aromatic compounds.

There are also several minor comments:

1. Last sentence of your abstract: Replace "enhances" with "improves" and delete "the" before "climate stability"

Response: Revised. It now reads, "This study improves our understanding of NOCs' contribution to $PM_{2.5}$ in TP and their potential impacts on climate stability in high-altitude areas."

2. Line 249, should be mechanisms.

Response: Revised. It now reads, "…despite no obvious direct local emission near the sampling site secondary formation of FAAs can occur through several mechanisms, including direct photolysis…".

3. Figure 3 (c), please use the same sizes for y-axis labels as panels (a-b)

Response: We revised the y-axis labels of Figure 3 (c), it now shows below:

[Figure]

**Figure 3** (a) Concentration and (b) composition of amines. (c) Temperature dependence of EA and MA, and (d) RH dependence of EA and MA. The box represents the 25th (bottom) and 75th percentiles (top), and the box-whisker data represents the 10th to 90th percentiles.